# Condition-Dependent Representational Alignment between Whisper and the Human Speech Network

**Chia-Chun Dan Hsu[1], [\*], Francis Pingfan Chien[2], [\*], Rong Chao[3], Ching Chih Sung[4], Yu-Te Wang[1], Po-Jang Hsieh[5], Yu Tsao[1]**

[1] Research Center for Information Technology Innovation, Academia Sinica, Taipei, Taiwan
[2] Taiwan International Graduate Program in Interdisciplinary Neuroscience,
National Taiwan University and Academia Sinica, Taipei, Taiwan
[3] Computer Science and Information Engineering, National Taiwan University, Taipei, Taiwan
[4] Graduate Institute of Communication Engineering, National Taiwan University, Taipei, Taiwan
[5] Department of Psychology, National Taiwan University, Taipei, Taiwan

## Abstract

Representations in modern speech models often align with human brain activity, but how acoustic degradation alters this alignment remains unclear. Here, we quantify condition-sensitive model–brain correspondence between an automatic speech recognition (ASR) model and human cortex. Twenty-five participants listened to clean and noisy ($-3$ dB SNR) sentences while undergoing fMRI. Layer-wise embeddings from Whisper Tiny (encoder–decoder Transformer) were mapped to voxel time series using ridge-regularized linear encoding to obtain normalized neural predictivity. Under clean speech, alignment peaked for decoder representations in the left middle frontal gyrus (MFG), with additional encoder peaks in the right inferior frontal gyrus (IFG). Under noisy speech, peaks shifted toward encoder layers in the right Heschl's gyrus and the right IFG pars orbitalis (IFGorb). Moreover, we observed significantly higher neural predictivity for clean than for noisy speech in the right IFG at middle and late encoder layers and in the left MFG at a middle decoder layer. These results demonstrate condition-dependent cortical alignment profiles across model layers and suggest a dynamic reweighting between feedforward acoustic encoding and top-down predictive decoding.

## 1   Introduction

Building models that mechanistically align with the brain requires mapping internal computations onto cortical anatomy and function, rather than merely predicting neural signals. Goal-driven modeling and integrative benchmarks have demonstrated that task-optimized networks can approximate sensory and language representations, motivating precise, layer-by-layer correspondences between model components and brain systems [1, 2]. In language processing, integrative modeling identifies predictive processing as a central organizing principle, emphasizing the importance of mappings that support functional ablations in neural networks and inform the development of robust brain–machine interfaces [3]. Directly optimizing models to predict fMRI responses has further clarified which architectural features increase brain-likeness [4]. Studies of auditory and language processing have shown that many models capture neural responses and reveal correspondences between computational stages and specific brain regions [5–7]. Research on layer alignment suggests that matching computational depth and dynamic properties, such as recurrence and attention mechanisms, improves correspondence with cortical processing [8, 9]. We extend this approach to speech, a domain in

---

[\*] Equal contribution.

Preprint.

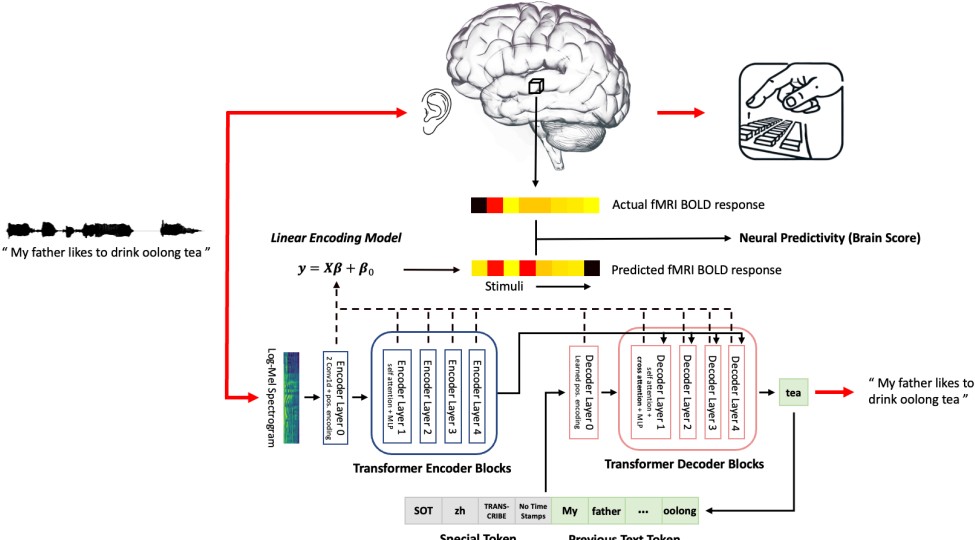

Figure 1: **Overall framework for aligning Whisper Tiny model layers with human brain activity and computing brain scores.** Speech stimuli are presented simultaneously to the Whisper Tiny ASR model and to human participants undergoing fMRI. Whisper Tiny yields a hierarchical set of representations across its encoder and decoder layers. For the same stimuli, we extract BOLD responses from the primary auditory cortex (Heschl's gyrus) and from bilateral language-network regions – the anterior temporal lobe (AntTemp), posterior temporal lobe (PostTemp), angular gyrus (AngG), inferior frontal gyrus (IFG), orbital IFG (IFGorb), and middle frontal gyrus (MFG). Layer-specific model embeddings are then used in encoding models to predict regional fMRI responses; the resulting neural predictivity (brain score) quantifies alignment between computational and neural representations.

which both the human cortex and Transformer-based ASR models exhibit hierarchical processing, progressing from acoustic analysis to lexical and semantic integration [10, 11]. We examine whether everyday challenges, such as background noise, reshape these correspondences. Specifically, by comparing clean and noisy speech, we test whether acoustic degradation shifts cortical alignment toward earlier layers associated with low-level auditory features, while clean speech enhances alignment with deeper layers that support predictive processing. This hypothesis is consistent with findings on clarity-dependent functional coupling in the language system [6, 12, 13]. In this work, we investigate the convergent alignment between fMRI signals and layer-wise embeddings from Whisper Tiny. We address the following questions: First, do latent representations in ASR models align with the representational structure observed in brain regions involved in speech processing under varying acoustic conditions? Second, which brain regions show the strongest convergence with ASR representations, and what does this reveal about the hierarchical organization of speech processing? Third, at what depth in the ASR model's layer hierarchy do its representations best correspond to brain responses, and how does this inform our understanding of speech comprehension stages? Finally, we quantify condition-specific differences in layer-level neural predictivity to examine shifts in representational alignment. Figure 1 outlines the full pipeline, from auditory stimuli to model representations to fMRI responses.

## 2 Methods

### 2.1 Participants & Stimuli & Experiment

We recruited 25 healthy native Mandarin speakers with normal hearing. Participants listened to 24 clean and 24 noisy Mandarin sentences (10 words; ∼3 each) presented via MRI-compatible headphones. Noisy trials were created by mixing each sentence with stationary speech-shaped noise (SSN) at −3 dB SNR [14]. After each sentence, while still in the scanner, they (i) rated intelligibility (how well did you comprehend this speech?), (ii) rated perceived speech quality (how would you

rate the quality of this speech?), and (iii) answered a two-option comprehension question about the sentence. Intelligibility and quality were reported on a 5-point mean opinion score (MOS) scale (1 = bad, 2 = poor, 3 = fair, 4 = good, 5 = excellent) [15]. Comprehension was scored binary as correct or incorrect.

## 2.2 fMRI Data & ROI Definition

MRI scanning was performed on a Siemens Magnetom Skyra 3T scanner. Anatomical images were acquired using a T1-weighted multi-echo MP-RAGE sequence (1 mm$^3$ voxels). Functional images were collected with a gradient-echo EPI sequence (TR = 2000 ms; TE = 24 ms; FOV = 220 × 220 mm; 38 slices; voxel = 3.4 × 3.4 × 4.0 mm ; flip angle = 90°). MRI preprocessing was performed in SPM12 (`http://www.fil.ion.ucl.ac.uk/spm/software/spm12`), including slice-timing correction, motion correction, co-registration of functional to anatomical images, normalization to MNI space, and smoothing with an 8 mm FWHM Gaussian kernel. First-level GLMs were specified with boxcar regressors for clean and noisy conditions, convolved with the Hemodynamic Response Function (HRF), plus six motion parameters as nuisance covariates. We employed a fixed set of subject-specific Fedorenko language fROIs: bilateral anterior temporal cortex (AntTemp), posterior temporal cortex (PostTemp), angular gyrus (Angular), inferior frontal gyrus (triangular and orbital subdivisions), and middle frontal gyrus (MFG) [16, 17]. Two additional auditory ROIs (bilateral Heschl's gyri) were defined anatomically using the Automated Anatomical Labeling (AAL) atlas, extracted via the MarsBaR toolbox [18].

## 2.3 ASR Model Representations

In this study, we evaluate the alignment relationship between brain and ASR model using layer-wise representations from Whisper-tiny [19]. In addition to Whisper-tiny's four transformer blocks in both encoder and decoder, we include the zeroth layer representation before each module. For each encoder layer $\ell$, Whisper-tiny generates 1500 hidden states for a 30-second input (each state representing $\sim$20ms), with dimensionality 384. Since our stimuli last approximately 3 seconds, we retain only the first 200 hidden states (corresponding to 4 seconds) to capture the entire stimuli duration, obtaining $\mathbf{E}^{(\ell)} \in \mathbb{R}^{200 \times 384}$. We then flatten to $\mathrm{vec}(\mathbf{E}^{(\ell)}) \in \mathbb{R}^{76800}$ and use this as the encoder layer-$\ell$ representation. The decoder in Whisper is an auto-regressive transformer that predicts the next token at each step, conditioned on the encoder's final representation and previously generated tokens. This objective encourages the pre-terminal hidden state to summarize the entire sentence information. Hence, we use the hidden state at the last non-special token position as the decoder layer-$\ell$ representation, $\mathbf{D}^{(\ell)} \in \mathbb{R}^{384}$.

## 2.4 Linear Encoding

We employ ridge regression as our linear encoding model to map ASR layer representations to BOLD responses across different ROIs and stimulus conditions. Ridge regression imposes constraints on the magnitude and covariance of regression weights to enhance weight estimation when dealing with limited and noisy data [20]. For implementation, we first reduce the encoder layer representation dimensionality from 76,800 to 2,724 using sparse random projection [21, 22]. This target dimensionality is determined by the Johnson-Lindenstrauss lemma with 24 samples and a distortion parameter $\epsilon = 0.1$ [23]. We apply scikit-learn's RidgeCV [24], selecting optimal regularization hyperparameters $\alpha$ for each voxel via nested leave-one-out cross-validation from 60 logarithmically spaced values between $10^{-6}$ and $10^6$. To evaluate the alignment relationship between ROIs and ASR model representations, we adopt the brain score metric [3]. We compute Pearson correlations between predicted and actual BOLD responses on a test set within the 4-fold cross-validation scheme. For each participant, we calculate the median predictivity across all voxels averaged over the 4 splits as the ROI raw score. Finally, we normalize this raw score by the noise ceiling, representing the theoretical upper bound of model performance given cross-subject consistency [3], yielding a brain score in the range [0, 1].

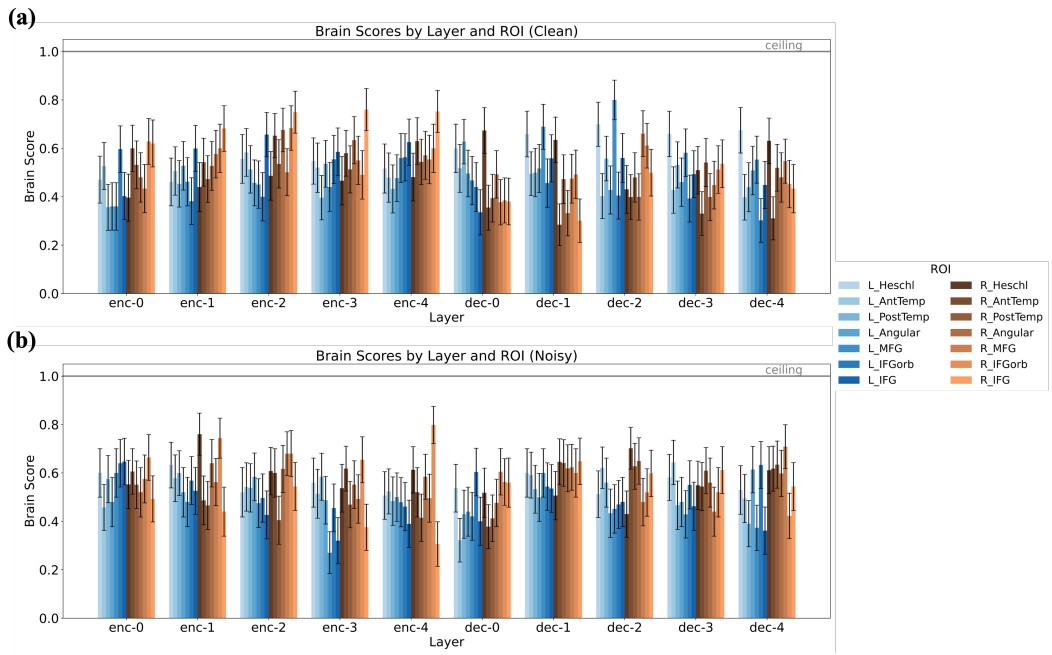

**(a)**

**(b)**

Figure 2: **Layer-wise alignment between Whisper-tiny representations and fMRI responses in 12 language- and auditory-related ROIs, quantified via normalized brain scores (predictivity) under clean and noisy speech. (a)** Clean speech: Alignment patterns across 5 encoder layers (enc-0 to enc-4) and 5 decoder layers (dec-0 to dec-4) for each ROI. **(b)** Noisy speech: Corresponding alignment patterns for the noisy ($-3$ dB SNR) condition. The horizontal grey line indicates the estimated noise ceiling (normalized to 1), representing the upper bound of model–brain predictivity given measurement noise. Error bars indicate $\pm 1$ SEM across 25 participants

## 3  Results

Normalized brain scores (BS; noise ceiling = 1.0) revealed orderly, layer-dependent alignment between Whisper Tiny and fMRI responses across 12 ROIs (Figure 2; Table 1). Under clean speech (Figure 2a), encoder predictivity increased with depth and reached its maximum at encoder layer 3 in Right IFG (BS = 0.76). Decoder alignment was highest at decoder layer 2 in Left MFG (BS = 0.80). Under noisy speech (Figure 2b), the layer profile was flatter and the strongest peaks were observed in auditory and orbitofrontal regions: encoder layer 1 in Right Heschl's gyrus (BS = 0.76) and encoder layer 4 in Right IFGorb (BS = 0.80). Consistent with this finding, Table 1 shows that, for clean speech, Right IFG was the best-performing ROI for encoder layers 1–4, and Left MFG/Left Heschl for decoder layers. For noisy speech, peak ROIs shifted toward Right IFGorb across multiple encoder layers and to MFG/AntTemp for decoder layers (typical peak $\approx 0.71$). To quantify the effect of acoustic degradation, we compared BS across conditions with paired tests and Benjamini–Hochberg FDR correction (Figure 3 and Figure 4). Clean speech showed significantly higher BS than noisy speech in Right IFG for encoder layers 3 and 4, and in Left MFG for decoder layer 2 ($q < 0.05$). Figure 3 plots clean in green and noisy in purple with the grey line marking the ceiling; error bars denote $\pm 1$ SEM ($n = 25$). Complementary behavioral measures collected in-scanner attested that intelligibility, perceived quality, and comprehension were significantly lower in the noisy condition (FDR-BH $p < 0.001$; Figure 5)

## 4  Discussion

Our data indicate that clean speech preferentially engages frontal language circuitry in a manner consistent with predictive, hierarchical processing. The strongest layer–ROI alignments under clear input localize to encoder layer 3, and layer 4 in the inferior frontal gyrus (IFG) and decoder layer 2 in the left middle frontal gyrus (MFG). Prior work places IFG at the center of hierarchical

Table 1: Best-performing ROI for each Whisper-tiny layer under clean and noisy speech conditions

| Layer | Clean Speech | | Noisy Speech | |
|---|---|---|---|---|
| | ROI | Brain Score | ROI | Brain Score |
| Encoder layer 0 | Right IFGorb | 0.63 | Right IFGorb | 0.66 |
| Encoder layer 1 | Right IFG | 0.68 | Right Heschl | 0.76 |
| Encoder layer 2 | Right IFG | 0.75 | Right IFGorb | 0.68 |
| Encoder layer 3 | Right IFG | 0.76 | Right IFGorb | 0.65 |
| Encoder layer 4 | Right IFG | 0.75 | Right IFGorb | 0.80 |
| Decoder layer 0 | Right Heschl | 0.67 | Right MFG | 0.60 |
| Decoder layer 1 | Left MFG | 0.69 | Right IFG | 0.65 |
| Decoder layer 2 | Left MFG | 0.80 | Right AntTemp | 0.70 |
| Decoder layer 3 | Left Heschl | 0.66 | Left AntTemp | 0.64 |
| Decoder layer 4 | Left Heschl | 0.67 | Right MFG | 0.71 |

structure building and the coordination of long-distance dependencies, providing a natural substrate for correspondence with mid/late encoder and mid-decoder representations that integrate context over time [25–27]. In parallel, naturalistic language studies link prediction during story listening to frontal systems, in line with our clean-speech peaks in IFG and MFG [12, 28, 29]. At the representational level, geometry-based analyses show that IFG brain embeddings share common structure with contextual language-model embeddings, enabling direct brain-to-model alignment [30]. This geometric concordance helps explain the maximal IFG alignment we observe with deeper model stages when the input is clean. Finally, the MFG peak at decoder layer 2 is compatible with its role within the multiple-demand (MD) network—a domain-general control system—suggesting that decoder computations leverage a frontal substrate recruited broadly for control and prediction when lexical and semantic information are strong [31]. By contrast, noise shifts alignment toward earlier acoustic stages and sensory and orbitofrontal regions. Under degraded input, peaks move to encoder layer 1 in right Heschl's gyrus and encoder layer 4 in right inferior frontal gyrus orbital part, while decoder-layer peaks weaken and become spatially dispersed. This pattern converges with evidence that speech-in-noise comprehension recruits a multisensory cortical network—including auditory, superior temporal, and parietal regions—to stabilize acoustic evidence and integrate complementary cues [32]. Functionally, the redistribution is consistent with a precision reweighting of computations: as the reliability of high-level predictions decreases, cortical processing emphasizes feed-forward analyses anchored in early auditory cortices, alongside evaluative and selection operations associated with orbitofrontal regions. Taken together, the layer–depth mapping we observe aligns with established cortical hierarchies. Deeper model stages (mid and late encoders, mid decoders) show maximal alignment in inferior and middle frontal regions under clean listening—coherent with IFG's and MFG's role in hierarchical sequence processing and integration with temporal cortex—whereas earlier model stages align with primary auditory cortex and neighboring regions, particularly under noise. Finally, human behavioral data (Figure 5) showed reduced intelligibility, perceived quality, and comprehension in the noisy condition—consistent with the observed neural dynamic reweighting and in line with predictive-processing findings.

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

# A  Appendix

## A.1  Model-Brain Alignment

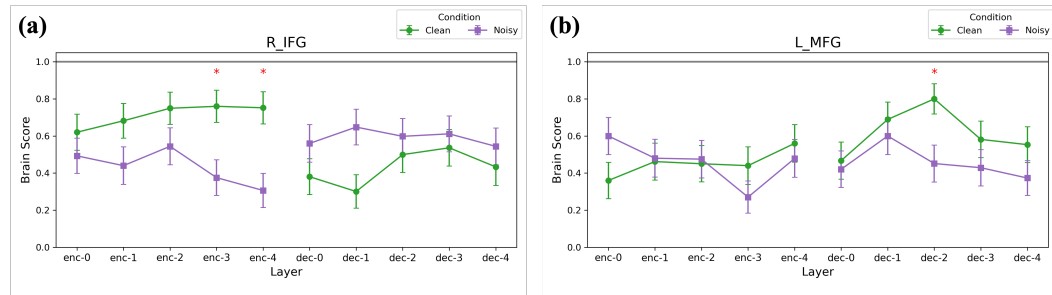

Figure 3: **Significant differences in model–brain alignment between clean and noisy speech conditions after FDR-BH correction. (a)** Right inferior frontal gyrus (R_IFG): Brain scores for clean (green) and noisy (purple) speech across encoder and decoder layers. Clean speech showed significantly higher alignment in encoder layers 3 and 4 (FDR-corrected $p < 0.05$, red asterisks). **(b)** Left middle frontal gyrus (L_MFG): Clean speech exhibited significantly higher alignment in decoder layer 2 (FDR-corrected $p < 0.05$, red asterisk). The horizontal grey line denotes the noise ceiling (normalized to 1). Error bars indicate ±1 SEM across 25 participants.

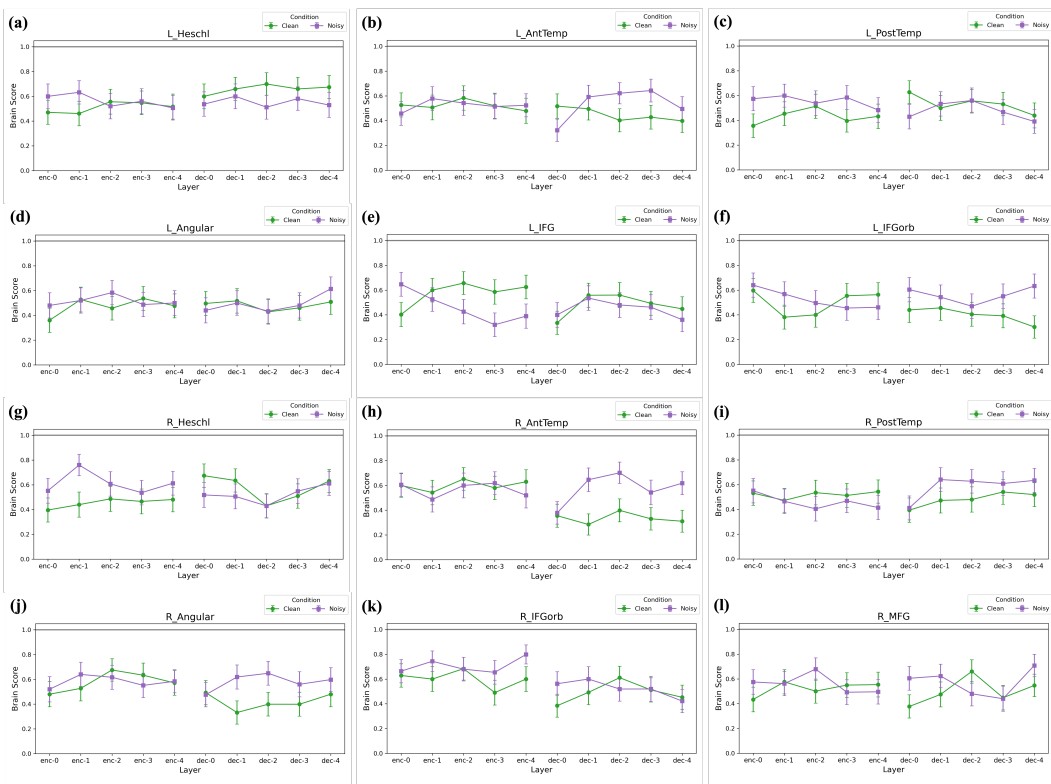

Figure 4: **Model–brain alignment between clean and noisy speech conditions.** Normalized brain scores (noise ceiling = 1.0) are plotted across Whisper-tiny layers for clean (green) and noisy (purple) conditions. The horizontal gray line indicates the noise ceiling. Error bars show ±1 SEM across 25 participants. Panels (a–l) correspond to: **(a)** Left Heschl, **(b)** Left AntTemp, **(c)** Left PostTemp, **(d)** Left Angular, **(e)** Left IFG, **(f)** Left IFGorb, **(g)** Right Heschl, **(h)** Right AntTemp, **(i)** Right PostTemp, **(j)** Right Angular, **(k)** Right IFGorb, **(l)** Right MFG.

## A.2 Behavior

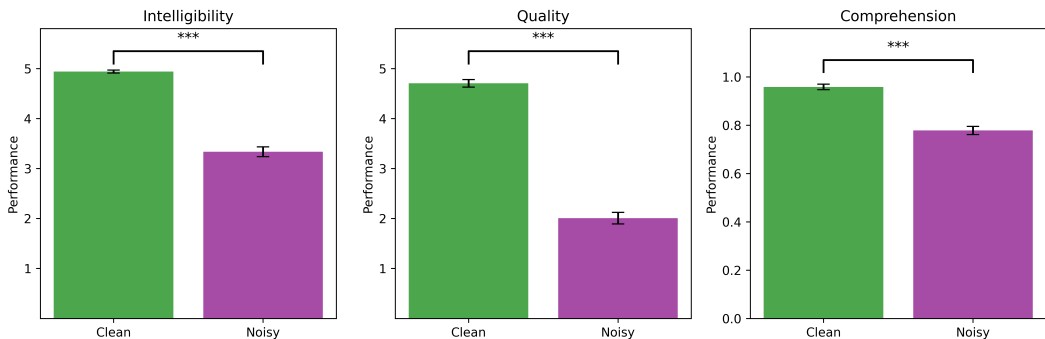

Figure 5: **Behavior under clean and noisy speech: Intelligibility (left), Quality (middle), and Comprehension (right).** Group means for intelligibility and perceived quality (MOS 1–5) and comprehension accuracy (0–1) while participants ($n = 25$) listened to 24 clean and 24 noisy ($-3$ dB SNR) Mandarin sentences during fMRI recording. Asterisks indicate significant clean–noisy differences after Benjamini–Hochberg FDR correction.

