# OpenReview forum: "Condition-Dependent Representational Alignment  between Whisper and the Human Speech Network"
_NeurIPS.cc/2025/Workshop/UniReps — UniReps2025_

### Official Review · Reviewer_XUah · 2025-09-12
**Representation alignment between human cortex and ASR models; interesting approach, details unclear**

**Confidence:** 2

**Review:**

Note: My expertise are in representational learning for text and I am not familiar with signal processing of fMRIs.

### Novelty

The paper discusses a straightforward approach to aligning representations of brain signals during speech recognition to a speech recognition transformer model, whisper tiny and  reports results under clean and noisy conditions. The novelty of this paper lies in reporting affects on alignment under noisy conditions.

### Strengths

The framework used is intuitive and experiment setup described in detail.

### Weakness

Paper lacks details and it is not transparent for following reasons:
1. Why this particular dataset was chosen, are their any public datasets available for this experiments?
2. Why were the two-option comprehension question framed as objective questions (answering by choosing)? The way these are designed could definitely induce bias.
3. Why was whisper tiny model chosen, can we experiment with other models as well.
4. Details on applications of this approach is missing. This would help NLP developers expand this approach to domains other than human cortex alignment to ASR.

**Score:**

2

**Topic Fit:**

2

---

### Official Review · Reviewer_Zxzb · 2025-09-13
**Condition-Dependent Representational Alignment between Whisper and the Human Speech Network**

**Confidence:** 3

**Review:**

This paper studies the alignment between representations from the Whisper-Tiny ASR model and human fMRI responses during speech comprehension, under clean and noisy acoustic conditions. The authors claim condition-dependent shifts in alignment: clean speech corresponds more strongly with higher-level frontal regions while noisy speech shifts peaks toward earlier auditory regions.


**Quality**: I have some reservations about the overall quality of the work. First, I think figure 2 could be easier to interpret (for example, may the authors could consider plotting the BS directly on the brain with ROI labels?). The brain score plots and ROI comparisons are dense could be better explained. The submission runs 4.5 pages when the limit is 4, and relies heavily on figures in the appendix. These figures are referenced without explicitly saying “in appendix,” which makes the main text feel incomplete and undermines self-containment.


The message that noise shifts alignment toward lower-level regions is plausible and consistent with prior work, but the experimental results here do not convincingly support this as a new or strong conclusion.


**Clarity**: The writing is generally fluent, but narrative clarity is weakened by overcomplicated figures and heavy reliance on appendix material. The key results are buried in figure panels and tables, rather than being clearly summarized in the text. The paper would benefit from simpler, more focused visualizations and a clear, strong, main message. I am also a bit confused by Figure 1. If I understand correctly, in the figure we see the linear encoding model being applied to encoder layer 0. However this is then repeated separately for every other layer, if I understand well. I think this should be specified in the caption for easier understanding. Lines 50-51: is it around 3 (seconds) each (sentence)? In the current form the text only says “10 words; ∼3 each” which unfortunately I find unclear.


**Originality**: the study of noise-dependent representational alignment is a meaningful addition to model-brain alignment work. However, the methodological approach (ridge regression encoding models, layer-wise ASR embeddings) is standard; the novelty lies mainly in contrasting clean vs. noisy speech.


**Significance**: In the current form, the contribution brings modest insights that are not strongly validated. If better validated and presented, this works would have high significance in neuroscience and speech technology/ASR.


**Overall**: the topic is relevant and the general direction is potentially very interesting but I feel the paper in its current form suffers from presentation issues and that the main claims are not adequately supported by the experiments. I am aware that the paper was presented to the extended-abstract track which encourages preliminary results, but I still feel like the current form of the work needs improvements.

I also believe that the excessive length of the paper with respect to the constraints merits consideration in the final decision.

**Score:**

2

**Topic Fit:**

3

---

### Official Review · Reviewer_vSnF · 2025-09-16
**Review: Alignment between the representations of an automatic speech recognition model and human brain activity when processing clean and noisy speech**

**Confidence:** 3

**Review:**

The paper studies the condition-dependent alignment between the layer-wise embeddings of Whisper Tiny, an automatic speech recognition model, and human fMRI signals while human participants listen to clean and noisy speech respectively inside an MRI scanner. They aim to investigate whether Whisper embeddings align with the structure of human brain signals for speech processing, and how this alignment shifts under varying acoustic conditions. For this they make 25 Mandarin speakers listen to clean and noisy Mandarin sentences who had to rate the intelligibility and perceived quality of the speech and answer a binary comprehension question. A ridge-regularised linear encoding model was used to predict BOLD signals across different brain ROIs and acoustic conditions using layer-wise Whisper encoder and decoder representations. Neural predictivity was measured as Pearson correlations between predicted and actual BOLD signals normalised by the noise ceiling. Results indicated that for clean speech, peaks in layer-ROI alignment occur in frontal language regions of the brain (encoder layers 3 and 4 in the IFG and decoder layer 2 in the left MFG), whereas for noisy speech, peaks shift towards earlier encoder layers and sensory and orbitofrontal regions (encoder layer 1 in right Heschl's gyrus and layer 4 in right IFG orbital and decoder layers become spatially dispersed). The findings are in line with established cortical hierarchies.

Strengths:
1. Provides analysis and evidence for acoustic degradation causing shift in representational alignment from top-down predictive decoding (under clean speech) to feed-forward acoustic encoding (under noisy speech), which is in alignment with prior established cortical hierarchies.
2. The condition-dependent layer-wise Whisper embeddings-ROI alignment analysis is a novel angle and has not been systematically studied in prior works.

Weaknesses:
1. Some technical details and design choices can be further elaborated on, such as the exact target signal that the linear encoding model predicts, whether it is a full time series, windowed response signals, or something else).
2. The impact of this work, specifically how the evidence for dynamic reweighting and predictive coding would help future model improvements or understanding of human speech processing should be analysed more thoroughly in the paper to highlight the significance of this work.

Quality: The preliminary results seem well-presented and comprehensive enough to provide a first set of evidences to further explore condition-dependent embeddings-ROI alignment.

Clarity: The presentation is mostly fine apart from scope of improvement in elaborating more on some technical details and design choices and the exact impact and applications of the findings.

Novelty: The systematic condition-dependent alignment analysis is a novel angle and has been missing in prior studies.

Significance: From my understanding, the work has potential to inspire model interpretability research in language and speech processing models to better align it with human perception in an attempt to increase efficiency and robustness. However, I believe that the authors should have explicitly mentioned some of these future directions and applications of their work. Regardless, the work is important from the perspective of understanding the internals of deep neural models.

Overall, given the Extended Abstract track, I would recommend the paper for acceptance.

**Score:**

4

**Topic Fit:**

3